# Induction of Oxidative Stress and Ferroptosis in Triple-Negative Breast Cancer Cells by Niclosamide via Blockade of the Function and Expression of SLC38A5 and SLC7A11

**DOI:** 10.3390/antiox13030291

**Published:** 2024-02-27

**Authors:** Marilyn Mathew, Sathish Sivaprakasam, Gunadharini Dharmalingam-Nandagopal, Souad R. Sennoune, Nhi T. Nguyen, Valeria Jaramillo-Martinez, Yangzom D. Bhutia, Vadivel Ganapathy

**Affiliations:** Department of Cell Biology and Biochemistry, Texas Tech University Health Sciences Center, Lubbock, TX 79430, USA; marilyn.mathew@ttuhsc.edu (M.M.); sathish.sivaprakasam@ttuhsc.edu (S.S.); gnandago@ttuhsc.edu (G.D.-N.); souad.sennoune@ttuhsc.edu (S.R.S.); nhi.t.nguyen@ttuhsc.edu (N.T.N.); valeria.jaramillo-martinez@ttuhsc.edu (V.J.-M.); yangzom.d.bhutia@ttuhsc.edu (Y.D.B.)

**Keywords:** SLC38A5, SLC7A11, TNBC cells, seleno-methionine, glutathione, glutathione peroxidase, Nrf2, ferroptosis, lipid peroxidation, niclosamide

## Abstract

The amino acid transporters SLC38A5 and SLC7A11 are upregulated in triple-negative breast cancer (TNBC). SLC38A5 transports glutamine, methionine, glycine and serine, and therefore activates mTOR signaling and induces epigenetic modifications. SLC7A11 transports cystine and increases the cellular levels of glutathione, which protects against oxidative stress and lipid peroxidation via glutathione peroxidase, a seleno (Se)-enzyme. The primary source of Se is dietary Se-methionine (Se-Met). Since SLC38A5 transports methionine, we examined its role in Se-Met uptake in TNBC cells. We found that SLC38A5 interacts with methionine and Se-Met with comparable affinity. We also examined the influence of Se-Met on Nrf2 in TNBC cells. Se-Met activated Nrf2 and induced the expression of Nrf2-target genes, including SLC7A11. Our previous work discovered niclosamide, an antiparasitic drug, as a potent inhibitor of SLC38A5. Here, we found SLC7A11 to be inhibited by niclosamide with an *IC*_50_ value in the range of 0.1–0.2 μM. In addition to the direct inhibition of SLC38A5 and SLC7A11, the pretreatment of TNBC cells with niclosamide reduced the expression of both transporters. Niclosamide decreased the glutathione levels, inhibited proliferation, suppressed GPX4 expression, increased lipid peroxidation, and induced ferroptosis in TNBC cells. It also significantly reduced the growth of the TNBC cell line MB231 in mouse xenografts.

## 1. Introduction

Iron is an obligatory element for the survival of all cells. It is involved in almost each and every biological process such as energy production, anabolism, catabolism, oxygen delivery, and the metabolism/disposal of drugs/xenobiotics. Iron exists in two valency states: ferrous (Fe^2+^) and ferric (Fe^3+^). It can be found in cells either in a free form (labile iron) or bound to proteins such as ferritin and transferrin. The two most important biologically active forms of iron are the iron–sulfur cluster (Fe-S) and heme. The transition between the two valency states is either enzyme-catalyzed (ferroxidases and ferrireductases) or simply the result of an electron transfer between iron and either electron acceptors (e.g., oxygen) or electron donors (e.g., vitamin C). The labile-free iron in the ferrous form is highly oxidative with reactivity towards hydrogen peroxide and lipid hydroperoxides via the Fenton reaction, resulting in the generation of hydroxyl and lipid alkoxyl radicals, which are extremely detrimental to cells, promoting a unique form of cell death known as ferroptosis [1,2,3]. 

Since many of the biological processes requiring iron are essential for cell proliferation, cancer cells are obligatorily programed to accumulate iron at levels more than that found in normal cells; in other words, cancer cells are “addicted” to iron [4,5]. This poses a conundrum: how can cancer cells accumulate an excess iron to support their growth and proliferation without risking themselves to ferroptotic cell death? Cancer cells overcome this problem by enhancing the efficacy of their antioxidant machinery. The most important component of this machinery is the glutathione (GSH)/glutathione peroxidase (GPX)/glutathione reductase (GR) system which involves the thiol-containing tripeptide glutathione (γ-glutamylcysteinylglycine), the selenium-containing enzyme GPX (particularly GPX4) and the reducing power NADPH. Cancer cells induce the cystine transporter SLC7A11 to increase the cellular levels of GSH [6,7,8], upregulate the hexose monophosphate shunt and malic enzyme to increase the production of NADPH [9,10,11] and increase the expression of GPX4 [12], which collectively detoxify hydroxyl and lipid alkoxyl radicals to protect the cells from ferroptosis despite the presence of excessive iron within the cells.

We undertook the present investigation to determine the potential crosstalk between two specific amino acid transporters in triple-negative breast cancer (TNBC) in enhancing the antioxidant machinery and to identify an effective pharmacologic strategy to interfere with this pathway as a plausible anticancer approach. SLC38A5 (also known as SN2 or SNAT5) is a transporter that mediates a Na^+^-dependent influx of glutamine, methionine, serine and glycine into cells in exchange for intracellular H^+^ [13,14]. It is upregulated in TNBC [15] and pancreatic cancer [16,17]. In addition to mediating the uptake of amino acids, SLC38A5 also promotes macropinocytosis, a special form of nutrient uptake in cancer cells [15,18,19]. In the present study with TNBC cells, we explored the possibility that SLC38A5 could transport seleno-methionine (Se-Met), the most predominant dietary source of selenium. Since the selenoenzyme GPX4 is a critical component of the antioxidant machinery in TNBC cells, the possible involvement of SLC38A5 in the delivery of selenium into cancer cells in the form of Se-Met could represent a novel function of the transporter in promoting cancer growth. Previously published studies have shown that Se-Met is an activator of Nrf2, an important transcription factor related to the antioxidant machinery [20]. The targets for Nrf2 include the cystine transporter SLC7A11 and the glutamate-cysteine ligase, the first enzyme in the synthesis of glutathione [21]. Similar to SLC38A5, SLC7A11 is also upregulated in TNBC [22,23]. Therefore, we hypothesized that there could be a functional coupling between SLC38A5 and SLC7A11 via Se-Met. We tested this hypothesis in the present study using TNBC cells. In addition, we have identified niclosamide, an FDA-approved antiparasitic drug, as a potent inhibitor of the function and expression of SLC38A5 and SLC7A11, consequently inducing oxidative stress, lipid peroxidation and ferroptotic cell death in TNBC cells and suppressing the growth of a TNBC cell line into tumors in mouse xenografts. Niclosamide is known to elicit anticancer effects via multiple mechanisms in several cancer types [24,25,26]. The findings of the present study provide yet another novel, hitherto unknown, pharmacological mechanism for the anticancer efficacy of this drug. 

## 2. Materials and Methods

### 2.1. Materials 

[2,3-^3^H]-L-Serine (specific radioactivity, 15 Ci/mmol) was purchased from Moravek, Inc. (Brea, CA, USA). [^3^H]-Glutamate (specific radioactivity 50.8 Ci/mmol) was purchased from PerkinElmer Corp (Waltham, MA, USA). Niclosamide, monomethyl fumarate, methionine, seleno-methionine and buthionine sulfoximine (BSO) were purchased from Millipore-Sigma (St. Louis, MO, USA). Ferrostatin-1, necrostatin-1, ZVAD-fmk and hydroxychloroquine were purchased from Santa Cruz Biotechnology Inc., (Dallas, TX, USA). 

### 2.2. Cell Lines and Culture Conditions 

We used three breast cancer cell lines (all representing triple-negative breast cancer). All media contained 10% fetal bovine serum. Cell cultures were tested every month for mycoplasma using a commercially available detection kit (cat. no. G238; Applied Biological Materials, Inc. Richmond, BC, Canada). All cell lines used in the present study were mycoplasma-free. Two of the breast cancer cell lines were from ATCC: MDA-MB231 (cat. no. CRM-HTB-26) and MDA-MB453 (cat. no. HTB-131) cells were cultured in Leibovitz’s L-15 medium. One patient-derived TNBC cell line, identified as TXBR100, was provided by the TTUHSC Cancer Center. This cell line was cultured in a special medium consisting of Dulbecco’s modified Eagle’s medium and Ham’s F12 medium, in a 1:1 ratio, supplemented with 20 ng/mL EGF, 0.01 mg/mL insulin, 500 ng/mL hydrocortisone and 100 ng/mL cholera toxin. 

### 2.3. Uptake Measurement 

Uptake of radiolabeled amino acids was used to monitor the transport function of SLC38A5. Since SLC38A5 is a Na^+^-coupled transporter with the involvement of H^+^ movement in the opposite direction, the uptake assays were done at pH 8.5 to create an outwardly directed H^+^-gradient across the plasma membrane to maximize the uptake activity. As there are several Na^+^-coupled amino acid transporters for serine, which was used as the substrate in most of the experiments in the present study, we cannot specifically monitor the function of SLC38A5 by using Na^+^-containing buffer. However, unlike other Na^+^-coupled transporters, SLC38A5 is tolerant to Li^+^ (i.e., SLC38A5 functions when Na^+^ is replaced with Li^+^). Therefore, we used an uptake buffer with LiCl in place of NaCl. The composition of the uptake buffer was 25 mM Tris/Hepes, pH 8.5, containing 140 mM LiCl, 5.4 mM KCl, 1.8 mM CaCl_2_, 0.8 mM MgSO_4_ and 5 mM glucose. Serine that was used as the substrate to monitor the transport function of SLC38A5 is also substrate for SLC7A5 (LAT1), which is a Na^+^-independent amino acid transporter; therefore, uptake of serine via this transporter will contribute to the total uptake measured in the LiCl-containing buffer, thus confounding the interpretation of SLC38A5-specific uptake. Therefore, we needed to suppress SLC7A5-mediated serine uptake while measuring the transport activity of SLC38A5. This was done by including 5 mM tryptophan in the uptake buffer to compete with and block the transport of serine mediated by SLC7A5; SLC38A5 does not transport tryptophan and, therefore, SLC38A5-mediated uptake will not be affected by tryptophan. To determine the contribution of diffusion to the total uptake of serine, the same uptake buffer but with LiCl replaced isosmotically with *N*-methyl-D-glucamine chloride (NMDGCl) was used. As such, the uptake was measured in two buffers: (i) LiCl-buffer, pH 8.5 with 5 mM tryptophan; (ii) NMDGCl-buffer, pH 8.5 with 5 mM tryptophan. The uptake in NMDGCl-buffer was subtracted from the uptake in LiCl-buffer to determine the transport activity of SLC38A5 [18]. 

SLC7A11 is a Na^+^-independent system that mediates the cellular entry of cystine in exchange for intracellular glutamate under physiological conditions. However, we routinely measure the activity of this transporter by cellular uptake of [^3^H]-glutamate under Na^+^-free conditions. Under these conditions, SLC7A11 mediates the cellular entry of [^3^H]-glutamate in exchange for intracellular unlabeled glutamate. The primary reason for not using radiolabeled cystine directly in uptake measurement of SLC7A11 transport activity includes difficulties such as insolubility of cystine, the availability of radiolabeled cystine from most commercial sources only in ^35^S-form with its relatively low half-life and the inability to determine the relative amounts of reduced and oxidized forms of cystine in the stock solution. Transport activity with L-[^3^H]-glutamate was measured using a Na^+^-free uptake buffer (25 mM Hepes/Tris, 140 mM *N*-methyl-D-glucamine chloride, 5.4 mM KCl, 1.8 mM CaCl_2_, 0.8 mM MgSO_4_ and 5 mM glucose, pH 7.5). Non-carrier-mediated uptake (i.e, diffusional component) was determined by measuring the uptake of [^3^H]-glutamate in the presence of excess unlabeled glutamate (5 mM). The transport activity of SLC7A11 was calculated by subtracting the diffusional component from total uptake.

Cells were seeded in 24-well culture plates (2 × 10^5^ cells/well) with the culture medium and were allowed to grow to confluency, which normally took 2 or 3 days depending on the cell line. Confluent cultures were used for uptake measurements. On the day of uptake measurement, the culture plates were kept in a water bath at 37 °C. The medium was aspirated and the cells were washed with uptake buffers. The uptake medium (250 μL) containing corresponding labeled amino acid as the tracer along with unlabeled glutamate, methionine and Se-Met (different concentrations for a dose–response study) were added to the cells. Following incubation for 15 or 30 min, the medium was removed and the cells were washed three times with ice-cold uptake buffer. The cells were then lysed in 1% sodium dodecyl sulfate/0.2 N NaOH and used for measurement of radioactivity. 

### 2.4. Quantitative RT-PCR 

Total RNA was extracted from cells using TRIzol Reagent (ThermoFisher Scientific, Waltham, MA, USA) and the RNA was reverse-transcribed using a high-capacity cDNA reverse transcription kit (ThermoFisher Scientific, Waltham, MA, USA) according to the manufacturer’s protocol. Quantitative RT-PCR was performed with Takara Taq Hot Start Version (TaKaRa Biotechnology, Shiga, Japan) or Power SYBR Green PCR master mix (Bio-Rad, Hercules, CA, USA). Primer sequences are shown in Appendix A. The relative mRNA expression was determined by the 2^−∆∆Ct^ method. Additionally, 18S was used as a housekeeping gene for normalization. 

### 2.5. Protein Isolation and Western Blot

Cells and tumor tissues were lysed in Pierce™ RIPA buffer (ThermoFisher Scientific, Waltham, MA, USA) supplemented with Halt™ Protease and Phosphatase Inhibitor Cocktail (ThermoFisher Scientific, Waltham, MA, USA). Homogenates were centrifuged, and supernatants were used for protein measurement via Pierce™ BCA Protein Assay Kit (ThermoFisher Scientific, Waltham, MA, USA). Nuclear and cytoplasmic protein extractions were performed using a commercial kit as per instructions in the manufacture’s protocol (#78833, ThermoFisher Scientific, Waltham, MA, USA), following treatment with methionine, Se-Met or monomethylfumarate. Western blot samples were prepared in Laemmli sample buffer (Bio-Rad Laboratories, Hercules, CA, USA). They were loaded onto a SDS–PAGE gel and transferred onto a PVDF membrane (Bio-Rad Laboratories, Hercules, CA, USA). The membrane was blocked and antibodies diluted in 5% nonfat dry milk (Bio-Rad Laboratories, Hercules, CA, USA) or in 5% bovine serum albumin (Irvine Scientific, Santa Ana, CA, USA) were used. Protein bands were visualized using Pierce™ ECL Western Blotting Substrate (ThermoFisher Scientific, Waltham, MA, USA) and developed on the autoradiography film (Santa Cruz, Dallas, TX, USA). Primary antibodies were purchased either from Cell Signaling (Danvers, MA, USA) [anti-GPX4 (#52455), anti-FTH (#4393), anti-HSP60 (#12165), anti-phospho-p70S6K (#9205), anti-p70S6K (#2708)] or from Abcam (Waltham, MA, USA) [Nrf2 (#ab62352)]. Secondary antibody Horseradish peroxidase-conjugated goat anti-rabbit IgG (#1706515) was purchased from Bio-Rad Laboratories (Hercules, CA, USA). For quantification of protein levels by the densitometric analysis, the experiment was carried out in triplicate and the data were collected from the resultant three Western blots.

### 2.6. Assays for Lipid Radicals (Ferroptosis) and Iron

Lipid radical (ferroptosis) assay and iron assay were performed as follows. Cells were cultured on a 25 mm glass coverslip until they reached 60–70% confluency (~48 h). At the time of the experiment, cells were washed with NaCl buffer, pH 7.5 and then incubated with 1 µM of LipiRadical Green (a lipid radical detection reagent, FDV-0042, Funakoshi, Tokyo, Japan) or Ferro-orange (an iron detection probe, F374, Dojindo, Rockville, MD, USA) in NaCl buffer, pH 7.5 for 20 min and then washed with NaCl buffer, pH 7.5. To analyze the effects of modifiers of iron levels and lipid peroxidation, the cells were co-treated with the modifiers and the respective fluorescent probe for 20 min and then washed. The glass coverslip containing the cells was then probed under an inverted microscope. The fluorescence imaging was captured using a Nikon T1-E microscope with A1 confocal super-resolution module (Nikon, Dallas, TX, USA), with a 60× objective, at 488 nm. The images represent a maximum projection intensity derived from a Z-stack. The fluorescence quantification was performed by measuring the corrected total cell fluorescence (CTCF) using Image J (version: 2.14.0/1.54f) and the following formula; CTCF = (integrated density) − (area of cell of interest) × (mean fluorescence of background).

### 2.7. Assay for Reactive Oxygen Species (ROS)

The probe of DCFH-DA was used to measure ROS. Briefly, cells were grown in a 96-well plate and then incubated with DCFH-DA (10 µM) at 37 °C for 30 min in dark. At the end of the incubation, cells were treated with different concentrations of niclosamide. Fluorescence intensity was monitored with a Microplate Reader (Glomax multi-detection system, Promega, Madison, WI, USA) at the excitation and emission wavelengths of 485 and 528 nm, respectively. Cellular fluorescence levels were expressed as % of control group (i.e., no treatment with niclosamide). 

### 2.8. Glutathione and Lipid Peroxidation Assay

Control and niclosamide-treated cells were used for measurement of cellular glutathione levels as instructed in the manufacturer’s protocol (GSH-Glo assay, Promega, Madison, WI, USA). The levels of malondialdehyde were measured using lipid peroxidation kit [Lipid Peroxidation (MDA) Assay Kit (MAK085), Millipore-sigma, St. Louis, MO, USA].

### 2.9. Colony Formation Assay

We performed the colony formation (clonogenic) assay with different doses of niclosamide on two different TNBC cell lines. Initial seeding was done with 500 cells/well and culture was continued for 10 days with culture medium replaced with fresh medium with freshly prepared niclosamide every other day. At the end of the 10-day time period, the medium was removed and the colonies were fixed with ice-cold methanol/acetone and then stained with Giemsa stain. After examination, lysis buffer was added (1% sodium dodecyl sulfate/0.2 N NaOH) and incubated in a shaker to extract the Giemsa stain and quantified using a Microplate Reader (Glomax multi-detection system, Promega, Madison, WI, USA).

### 2.10. MTT Assay

Cells were seeded in 96-well plates; after 24 h, niclosamide treatment was initiated. Cells were then cultured for 72 h with fresh medium containing freshly prepared niclosamide supplied every 24 h. Cells were washed with phosphate-buffered saline twice followed by MTT reagent (ATCC). Treatment and lysis of the cells were done as per the manufacturer’s instructions. Absorbance of the lysate was measured at 550 nm. Cell viability assay with various cell-death inhibitors was also performed as described above. Cells were treated with niclosamide (1 µM) along with cell death inhibitors (10 µM), except hydroxychloroquine (25 µM) for 48 h.

### 2.11. Cell Invasion Assay

The effect of niclosamide on cell invasion was monitored using Corning^®^ BioCoat™ Matrigel^®^ Invasion assay kit according to the manufacturer’s instructions. Briefly, cells were serum-starved and their invasion to the other side of the membrane was analyzed in the presence or absence of niclosamide for 24 h. At the end of this treatment, non-invaded cells on the top side of the membrane were removed by scrubbing. Cells which invaded the other side of the membrane were fixed with 100% methanol, stained with crystal violet and counted under an inverted microscope; images were also captured.

### 2.12. Mouse Xenograft Experiments

Female athymic nude mice (4-week-old) were purchased from Jackson Laboratories and housed under standard conditions. MDA-MB231 cells were injected into lower mammary fat pad (5 × 10^6^ cells). All cells were suspended in serum-free media and Matrigel (1:1 ratio), with 100 µL of suspension being injected into each mouse. Mice were treated by daily intraperitoneal injection of niclosamide (4 mg/kg/day) and vehicle (dimethylsulfoxide) control. The treatment began when the tumor size was 100–150 mm^3^. Tumor size was measured biweekly with a caliper, with tumor volume calculated using the formula (width^2^ × length)/2. Tumors were allowed to grow for 7 weeks; mice were then euthanized via isoflurane injection and tumors harvested. The animal study protocol was approved by the Texas Tech University Health Sciences Center Institutional Animal Care and Use Committee (IACUC protocol number 18005) and the experiments were conducted in the same institution. RNA and protein were prepared from the tumor tissue for qRT-PCR and Western blotting. 

### 2.13. Homology Modeling and Docking Studies 

SLC38A5 homology modeling and docking studies were performed as previously published [18] to determine the theoretical values for the binding energies for the interaction of methionine and seleno-methionine with SLC38A5. Since the cryo-EM crystal structure of human SLC38A5 is not known, we used the structures of closely related transporters as templates for our purpose [18]. With regard to structural modeling of the cystine/glutamate antiporter, we have information on the cryo-EM crystal structure of human transporter–chaperone complex SLC7A11/SLC3A2 (PDB: 7P9V) [27]. This structure of the in vivo functional heterodimer was used for docking studies to determine the binding energies for methionine and selenomethionine. In the case of both transporters, the docking simulations were conducted using AutoDock/Vina in conjunction with the USCF Chimera program [28,29]. A grid with dimensions of 30 × 30 × 30 (Å3) was employed, focusing on the cavity within the structures of the two proteins where ligand binding was observed. 

### 2.14. Statistics 

Uptake experiments were routinely done in triplicates and each experiment was repeated at least thrice using independent cell cultures. Statistical analysis was performed with a two- tailed, paired Student’s *t*-test for single comparison and a *p*-value < 0.05 was considered statistically significant. Data are given as means ± S.E. For quantification of fluorescence signals in image analysis related to ferroptosis, ANOVA followed by Dunn’s test was used to determine the significance of difference among the different groups.

## 3. Results

### 3.1. Expression Patterns for SLC38 Gene Family Members in Breast Cancer

The SLC38 gene family consists of 11 members and the best characterized among them in terms of transport function are five members, all belonging to two subclasses: system A (SLC38A1, SLC38A2 and SLC38A4) and system N (SLC38A3 and SLC38A5) [30]. There is evidence for a role for two of these transporters in TNBC: SLC38A2 [31] and SLC38A5 [15,18]. Using the transcriptomic data from publicly available datasets, we analyzed the expression of the remaining three members in breast cancer (Appendix A). This analysis showed that SLC38A1 is upregulated in estrogen receptor-positive luminal-type breast cancer. In contrast to SLC38A1, SLC38A3 is downregulated in this type, whereas SLC38A4 remains unaltered. None of the three are altered in HER2-positive breast cancer. In TNBC, SLC38A1 remains unaltered, SLC38A3 is upregulated and SLC38A4 is downregulated. 

### 3.2. Interaction of Seleno-Methionine (Se-Met) with Human SLC38A5

There is no information in the published literature on transport systems available for Se-Met in mammalian cells. However, it is likely that any transport system that is capable of transporting methionine might transport Se-Met because the only structural difference between methionine and Se-Met is the replacement of sulfur in methionine with Se. Therefore, we hypothesized that the SLC38A5 that transports methionine would be able to recognize Se-Met as a substrate. Since radiolabeled Se-Met is not available from any commercial source, we decided to address this issue by monitoring the ability of Se-Met to compete with serine for uptake that is mediated by SLC38A5. The serine uptake was measured in two different TNBC cell lines (MB231 and TXBR100). The uptake conditions were designed such that only the SLC38A5-mediated uptake of serine was measured, as described in the Section 2. The abilities of methionine and Se-Met as competitors of serine uptake was compared in a dose–response experiment in both cell lines (Figure 1). Methionine as well as Se-Met competed with serine for SLC38A5-mediated transport. In MB231 cells, the *IC*_50_ values for the inhibition of serine uptake by methionine and Se-Met were 518 μM and 680 μM, respectively (Figure 1A,C). A similar inhibitory potency was observed in the TXBR100 cell line also (*IC*_50_ values: 316 μM for methionine and 453 μM for Se-Met) (Figure 1B,D). Since the concentration of serine in these uptake experiments was only 5 μM, very small compared to its *K*_t_ value for SLC38A5-mediated transport (200–1300 μM) [13,32], the *IC*_50_ values for methionine and Se-Met can be approximated to their corresponding *K*_t_ values. 

We also analyzed the interaction of methionine and Se-Met with human SLC38A5 using a theoretical approach. We modeled the interaction of these two amino acids with the transporter by molecular docking, as we described in our previous publication where we used a similar approach to identify the FDA-approved drug niclosamide as a ligand for the transporter [18]. With this approach, we determined that methionine interacted with SLC38A5 with a binding energy of −4.8 kcal/mol and that Se-Met also interacted with the transporter with a similar binding energy (−4.9 kcal/mol). These binding energies translate into corresponding dissociation constants (*K*_D_) of 295 μM for methionine and 250 μM for Se-Met. As the dissociation constant is an approximate equivalent of *K*_t_ values, these theoretically determined values are in the same range as the experimentally determined values shown in Figure 1. In addition to the similar binding affinities between methionine and Se-Met for SLC38A5, the amino acid residues in SLC38A5 that are responsible for the binding of methionine and Se-Met are also the same (Y142, L194, T197, S198, F267, A268, E274 and T349). However, the interaction of Se-Met showed the participation of two additional amino acid residues (H134 and N135). 

### 3.3. Potentiation of Nrf2 and Its Antioxidant Signaling in TNBC Cells by Se-Met

Se-Met is the primary dietary source of selenium [33] and the sole biological function of this micronutrient is its involvement in the catalytic activities of selenoproteins such as glutathione peroxidases which are components of cellular antioxidant machinery [34]. However, Se-Met also potentiates the antioxidant machinery in mammalian cells through a second mechanism by enhancing the expression and activity of Nrf2, an important antioxidant transcription factor [20]. Since TNBC cells present the robust expression of SLC38A5 and also since SLC38A5 appears to mediate the cellular entry of Se-Met, we asked if Se-Met would potentiate the antioxidant machinery in TNBC cells by enhancing Nrf2 signaling. To address this question, we exposed two different TNBC cell lines (MB231 and TXBR100) to Se-Met (250 μM and 1 mM) for 24 h and then examined the expression and subcellular localization of Nrf2 by immunostaining. These studies showed that the exposure of the cells to Se-Met increased the cellular levels as well as the nuclear localization of Nrf2 in both cell lines (Figure 2A,C, MB231; Figure 2B,D, TXBR100). The effect was noticeable even at 250 μM. In these experiments, we used monomethylfumarate (0.1 mM) as a positive control because this compound is a known activator of Nrf2 signaling [35]. Monomethylfumarate increased the cellular levels and the nuclear localization of Nrf2 (Figure 2). The nuclear translocation of Nrf2 in response to treatment with Se-Met and monomethylfumarate was additionally confirmed by Western blot with appropriate internal controls (vinculin for cytosol and lamin B1 for nucleus) after the separation of cytosol and nuclear fractions (Figure 2E,F). Methionine did not have any noticeable effect on Nrf2 expression or on Nrf2 nuclear translocation.

Nrf2 is a transcription factor and its antioxidant activity is evident from the biological activities of its transcriptional targets, which include the cystine transporter SLC7A11, the heme-degrading enzyme heme oxygenase-1 (HO-1) and the components of the glutathione synthetic pathway (the catalytic and the modulatory subunits of glutamate–cysteine ligase (GCLC and GCLM) that catalyzes the first step in the synthesis of glutathione) [22]. To determine if the observed increase in the cellular and nuclear levels of Nrf2 seen in Se-Met-exposed TNBC cells is accompanied with a parallel increase in the expression of the above-mentioned Nrf2 target genes, we compared the levels of mRNA for SLC7A11, HO-1, GCLC and GCLM in control and Se-Met-treated MB231, TXBR100 and MB453 cells by qRT-PCR (Figure 3A–C). We found a significant increase in mRNA levels for all four targets in response to Se-Met treatment in all three TNBC cell lines.

To corroborate the data on the Se-Met-induced changes in mRNA levels with functional data, we selected SLC7A11 for examination. We measured the transport activity of SLC7A11 in control cells and in cells treated with Se-Met. We used monomethylfumarate as a positive control. In addition, we compared the effects of Se-Met with that of methionine to determine if the observed effects are specific for Se-Met. The transport activity of SLC7A11, monitored as the glutamate uptake, as detailed in the Section 2, increased substantially in Se-Met-treated cells (all three TNBC cell lines: MB231, TXBR100 and MB453) (Figure 3D–F). The positive control monomethylfumarate also elicited the stimulatory effect on SLC7A11 transport activity. Importantly, methionine did not have any effect, indicating that the potentiating effect on Nrf2 signaling in TNBC cells is specific for Se-Met and that the effect is not seen with methionine.

### 3.4. Identification of Niclosamide as a Potent Blocker of SLC7A11

We reported recently that the FDA-approved antihelminthic drug niclosamide is a potent inhibitor of SLC38A5- and SLC38A5-mediated macropinocytosis [18]. This study first involved a theoretical molecular docking approach, followed by actual experimentation in mammalian cells. We followed a similar strategy to determine if niclosamide has any effect on SLC7A11. First, we analyzed the binding of niclosamide with human SLC7A11 using the recently elucidated cryo-EM structure of this transporter (Section 2). We modeled the binding of niclosamide with SLC7A11 and compared it with the binding of erastin, the most potent blocker of SLC7A11 known to date in the published literature [36]. The results, shown in Appendix A, demonstrate that both compounds interacted with the transporter even though the amino acid residues involved in the binding are different for both compounds, as could be expected based on the differences in their chemical structures. The binding energy obtained from this approach was −8.0 kcal/mol for niclosamide; the corresponding value for erastin was −7.3 kcal/mol. We also did the docking for the influx substrate cystine and the exchange substrate glutamate for the transporter; the binding-energies were −4.8 and −4.7 kcal/mol, respectively. The theoretical *K*_D_ values derived from these binding energy values are 1.3 μM for niclosamide, 4.3 μM for erastin, 300 μM for cystine and 350 μM for glutamate. 

We then examined experimentally the ability of niclosamide to inhibit SLC7A11 transport activity in TNBC cells by using the glutamate uptake as the readout (Figure 4A–C). In three different TNBC cell lines, niclosamide inhibited the uptake with an *IC*_50_ value of ~0.3 μM, an affinity at least three-to-four times greater than the experimental value known for erastin [36,37]. Due to the significant difference between the experimental (~0.3 μM) and theoretical (1.3 μM) values for niclosamide affinity, we compared the experimental and theoretical values for the affinities of the two substrates (cystine and glutamate) and five known inhibitors of this transporter (erastin, sulfasalazine, sorafenib, S-4-carboxy phenylglycine along with niclosamide) to see if a similar trend exists for the other substrates and inhibitors as well (Figure 4D). The experimental and theoretical values correlated well for all of them (r^2^ = 0.9; *p* < 0.001), thereby providing credibility to the potential of this theoretical approach to discover new inhibitors, as we demonstrated with niclosamide.

### 3.5. Influence of Niclosamide on Nrf2 Expression in TNBC Cells

Niclosamide targets multiple signaling pathways in mammalian cells; these pathways include those involving Wnt, STAT3, mTOR and NOTCH [38,39,40]. Since we found in the present study that niclosamide blocks the transport function of the cystine transporter SLC7A11, a key component of the cellular antioxidant machinery, we asked if the treatment of TNBC cells with this drug would have any impact on the expression of Nrf2, a key transcription factor that protects the cells against oxidative stress. The treatment of MB231 cells with niclosamide (1–2.5 μM) for a short time (4 h) did not have any noticeable effect on Nrf2 levels or its subcellular localization as assessed by immunostaining. However, when the treatment time was extended to 24 h, niclosamide (2 μM) decreased the levels of Nrf2 mRNA (qRT-PCR) and protein (Western blot) (Figure 5A). The decrease in Nrf2 protein was evident also in immunofluorescence analysis (Figure 5B). In TXBR100 cells, however, 24 h treatment with niclosamide (2 μM) did not alter the levels of Nrf2 mRNA, but significantly decreased the protein level (Figure 5C).

### 3.6. Induction of Ferroptosis by Niclosamide in TNBC Cells

The data presented thus far above show that SLC38A5, which is upregulated in TNBC, could provide selenium to cancer cells via the mediation of the cellular uptake of Se-Met and that Se-Met potentiates Nrf2 signaling with a resultant increase in the expression and function of SLC7A11 and glutathione-synthesizing machinery. The data also show that the FDA-approved drug niclosamide blocks the transport function of SLC38A5 and SLC7A11 and also decreases the cellular levels of the antioxidant transcription factor Nrf2. This suggests that niclosamide is capable of inducing oxidative stress in cancer cells by decreasing the cellular levels of glutathione. Cancer cells are “addicted” to iron as iron is needed for multiple biochemical pathways that are closely associated with cell proliferation and growth [4,41]. But excess free “labile” iron is detrimental to cells because of its ability to generate reactive oxygen species via the Fenton reaction and cause the breakdown of polyunsaturated fatty acids present in the form of phospholipids in biological membranes, a process known as lipid peroxidation. This pathway leads to a unique form of cell death, called ferroptosis. As such, cancer cells need to accumulate excess iron to support cell proliferation, but at the same time have to find ways to protect themselves from ferroptotic cell death. They manage to do this by upregulating their antioxidant machinery. Since our studies have now shown that niclosamide interferes with this protective mechanism against ferroptosis, we asked whether niclosamide renders TNBC cells susceptible to ferroptotic cell death. We addressed this question with two different TNBC cells (MB231 and TXBR100). Ferroptosis was monitored using a fluorescent probe that is selective for byproducts of lipid peroxidation, as described in the Section 2. We found that niclosamide at low micromolar concentrations (5 μM) induced ferroptosis in both cell lines (Figure 6). The observed increase in fluorescent signals in response to treatment with niclosamide was indeed due to ferroptosis because ferrostatin 1, a specific inhibitor of ferroptosis, blocked this increase, whereas necrostatin (an inhibitor of necroptosis) and ZVAD-fmk (an inhibitor of apoptosis) did not have any effect (Figure 6).

However, we noted a significant difference between MB231 cells and TXBR100 cells, the former being more sensitive to niclosamide-induced ferroptosis than the latter. In MB231 cells, 30 min of exposure to the drug was sufficient to induce marked ferroptosis, whereas there was a much lower magnitude of ferroptosis induction in TXBR100 cells under identical experimental conditions. However, when the exposure time to the drug was increased from 30 min to 2 h, we saw a much higher magnitude of ferroptosis induction even in TXBR100 cells. 

### 3.7. Biochemical Consequences of Niclosamide Treatment in TNBC Cells Relevant to Ferroptosis Induction

To understand the molecular basis of the observed induction of ferroptotic cell death by niclosamide, we first monitored the levels of glutathione in the control and niclosamide-treated TNBC cells. MB231 cells and TXBR100 cells were treated with 1 or 2 μM niclosamide for 24 h and then cell lysates were prepared for the determination of the glutathione levels. We used buthionine sulfoximine as a positive control; this compound is a widely used potent inhibitor of glutathione synthesis. In both cell lines, treatment with niclosamide led to a marked decrease in cellular levels of glutathione (Figure 7A,B). As expected, buthionine sulfoximine (100 μM) caused ~90% inhibition in glutathione in both cell lines. In comparison, the decrease in glutathione levels was 60–70% when the cells were treated with 2 μM niclosamide. These data, showing a decrease in glutathione in response to niclosamide treatment, corroborate with the findings that niclosamide blocks the transport activity of SLC38A5 and SLC7A11 and also suppresses Nrf2 signaling with the resultant decrease in the cysteine supply and glutathione synthesis.

Since niclosamide interferes with multiple signaling pathways [38,39,40], we asked if the drug is capable of affecting the expression of SLC38A5 and SLC7A11 in addition to its ability to block the function of these two transporters by its direct interaction with the transporters. To address this question, we monitored the levels of mRNA for these two transporters in the control and niclosamide-treated TNBC cells. We found that niclosamide decreased the levels of SLC38A5 mRNA and SLC7A11 mRNA to a marked extent in both cell lines (Figure 7C,D). The effect on SLC38A5 mRNA was much more in magnitude than the effect on SLC7A11 mRNA. These data show that niclosamide elicits a negative impact on these two transporters by two distinct mechanisms: (i) it directly interacts with the transporters and blocks their transport function and (ii) it also suppresses the expression of these two transporters.

Since niclosamide treatment induced ferroptosis, we monitored the impact of the drug exposure on the cellular levels of glutathione peroxidase 4 (the enzyme that removes hydrogen peroxide and lipid peroxides) and ferritin (the storage protein for iron that reduces the cellular levels of free “labile” iron). In MB231 cells as well as in TXBR100 cells, niclosamide treatment decreased glutathione peroxidase 4 protein levels (Figure 7E,F). Ferritin levels also were lower in niclosamide-treated cells than in control cells, but the effect was much more pronounced in TXBR100 cells than in MB231 cells (Figure 7F).

Niclosamide is known to suppress mTOR signaling [38,39,40]. Hence, we examined the effect of niclosamide on this pathway in our experimental system. We found a profound decrease in the levels of p-S6K and phospho-p-S6K, the downstream components in the pathway, in both TNBC cell lines in response to niclosamide treatment (Figure 7G).

We also monitored the levels of iron and reactive oxygen species (ROS) as well as malondialdehyde (a marker for lipid peroxidation) in TNBC cells in response to niclosamide treatment. The iron levels increased in both cells upon niclosamide treatment (Figure 8A–C). In this experiment, we used exposure to ferric ammonium citrate as a positive control to increase the cellular levels of iron. We also found an increase in ROS levels (Figure 8D) and malondialdehyde levels (Figure 8E) in niclosamide-treated cells compared to control cells.

Taken collectively, these data demonstrate that the exposure of TNBC cells to niclosamide increases the cellular levels of iron and ROS, decreases glutathione and GPX4 levels and the expression of SLC38A5 and SLC7A11 and increases lipid peroxidation, all contributing to the observed induction of ferroptosis. The data also provide evidence for the suppression of the mTOR signaling pathway by niclosamide. 

### 3.8. Effect of Niclosamide on Colony Formation and Cell Proliferation in TNBC Cells

We assessed the impact of niclosamide treatment on the proliferation and colony formation ability of TNBC cells (MB231 and TXBR100) in vitro. The exposure of the cells to niclosamide had a drastic inhibitory effect on their colony formation (>50% decrease at 0.5 μM) (Figure 9A,B) and cell proliferation (>50% decrease at 2 μM) (Figure 9C,D). Even with 0.25 μM niclosamide, a significant inhibitory effect was noticeable in the colony formation assay, though the magnitude of the effect was lower at this concentration compared to the effect observed with 0.5 μM niclosamide. The inability of the cells to grow in the presence of niclosamide was principally due to the drug-induced ferroptosis, as evident from the findings that only ferrostatin 1 improved cell viability in the presence of niclosamide (Figure 9E,F). The inhibitors of necroptosis (necrostatin), apoptosis (ZVAD-fmk) and autophagic cell death (hydroxychloroquine) failed to promote cell viability in the presence of niclosamide.

We also monitored the influence of niclosamide on the invasion and migration of TNBC cells when exposed to niclosamide (2.5 and 5 μM). At both concentrations, niclosamide had a marked suppressive influence on the invasion/migration capacity of both TNBC cell lines (Figure 10). Even at 2.5 μM, the inhibition of cell invasion and migration by niclosamide was greater than 75%. 

### 3.9. Effect of Niclosamide on the Growth of a TNBC Cell Line into Tumor When Xenografted in Mice

The in vitro experiments described above suggested the potent tumor-suppressive activity of niclosamide by the ability of the drug to induce oxidative stress and iron-induced ferroptosis via the blockade of the expression and function of SLC38A5 and SLC7A11. Therefore, we examined the anticancer efficacy of this drug in vivo using the mouse xenograft model with the TNBC cell line MB231. We initiated drug administration when the tumors grew to a size of ~100–150 mm^3^. The drug was given daily by an intraperitoneal injection at a dose of 4 mg/kg of body weight. We found a drastic reduction in the growth of the tumors in drug-exposed mice (Figure 11A–C). At the end of the experimental period, the tumors were harvested and prepared for qRT-PCR and Western blot. The levels of mRNAs for Nrf2, SLC7A11 and SLC38A5 were markedly decreased in niclosamide-exposed tumors in comparison with control tumors (Appendix A). Western blot analysis showed that Nrf2 protein levels were also markedly decreased in niclosamide-exposed tumors in comparison with control tumors (Figure 11 D,E). The protein levels of the ferritin heavy chain also decreased (Figure 11 D,E) even though the magnitude of the decrease was smaller than that for Nrf2. These findings provide convincing evidence for the in vivo efficacy of niclosamide as an anticancer drug for TNBC and corroborate the in vivo evidence for the molecular aspects of the drug’s anticancer effects which were observed in vitro.

## 4. Discussion

Among the five members of the SLC38 gene family which have been characterized in detail as plasma membrane amino acid transporters, SLC38A5 and SLC38A3 are upregulated in TNBC. SLC38A2 and SLC38A4 are downregulated and SLC38A1 remains unaltered. We have recently shown that SLC38A5 functions as a tumor promoter in TNBC [15,18]. SLC38A2 has also been shown to promote TNBC [31], even though its expression is downregulated [15]. We have no information on the potential role of SLC38A3 which is upregulated and that of SLC38A4 which is downregulated. SLC38A3 and SLC38A5 possess almost identical functional features [30] and, therefore, it is likely that this transporter also promotes the growth and proliferation of TNBC cells. On the other hand, SLC38A4 has unique functional features such as the ability to transport not only neutral amino acids, but also cationic amino acids. It is also an imprinted gene [42] and has been shown to function as a tumor suppressor in some cancers [43]. Even though the downregulation of the transporter in TNBC may suggest a similar role in this cancer, it has not yet been determined experimentally.

Previous studies from our lab have uncovered an unconventional function of SLC38A5 [18]. Its transport function as an amino acid-dependent Na^+^/H^+^ exchanger couples amino acid entry into cells via this transporter to intracellular alkalinization, which promotes macropinocytosis. Since SLC38A3 functions in an identical manner in terms of transport modality, we have postulated that this transporter might also promote micropinocytosis, even though it is only speculative at this time and has not yet been validated experimentally [19]. In the present study, we have uncovered another important functional feature of SLC38A5 in TNBC. It plays a role in selenium nutrition in TNBC cells by its ability to deliver Se-Met into cells. Selenium is obligatory for the antioxidant function in mammalian cells and, therefore, the SLC38A5-mediated delivery of selenium ought to be an essential feature of this transporter as a tumor promoter in TNBC. In addition to the role in selenium nutrition, the SLC38A5-mediated delivery of Se-Met plays a role in the control of Nrf2 signaling. Since SLC7A11 is an important target of Nrf2-mediated transcriptional activity, this suggests a functional coupling between SLC38A5 and SLC7A11 with Se-Met as an intermediate. SLC7A11 has already been shown to function as a tumor promoter in TNBC [22,23]. Therefore, the ability of SLC38A5 to potentiate Nrf2 signaling with a resultant increase in SLC7A11 expression is important, underscoring the tumor-promoter role of SLC38A5.

Based on the data presented in this paper, we conclude that there are two aspects relating to the antioxidant function of SLC38A5 in TNBC cells. First, the function of the transporter has a direct positive effect on the antioxidant machinery of tumor cells by maintaining the optimal selenium nutrition via the delivery of Se-Met. This micronutrient is obligatory for the function of glutathione peroxidases. Second, the SLC38A5-mediated delivery of Se-Met into tumor cells potentiates the activity of the transcription factor Nrf2, which not only increases the ability of the tumor cells to synthesize glutathione by upregulating the expression of GCLC and GCLM to promote the first step in the glutathione synthetic pathway, but also induces the expression of the transporter SLC7A11 which provides cysteine (in the form of cystine), the rate-limiting amino acid for glutathione synthesis in tumor cells. Thus, Se-Met enables the functional coupling between SLC38A5 and SLC7A11 and this crosstalk between the two transporters forms an integral part of the antioxidant machinery in tumor cells.

The present study has also explored the impact of the FDA-approved drug niclosamide on the antioxidant machinery of TNBC cells. We have already shown that niclosamide is a potent inhibitor of SLC38A5 [18]. Therefore, we could speculate that the ability of TNBC cells to acquire Se-Met via SLC38A5 would be impaired when exposed to niclosamide, thus decreasing the cellular levels of Se and, hence, the catalytic activity of glutathione peroxidases. The experiments described in the present study have discovered another important action of niclosamide that is related to the antioxidant machinery of TNBC cells. This drug is also a potent inhibitor of SLC7A11. In fact, the potency of the inhibition observed identifies niclosamide as the most potent inhibitor of SLC7A11 known to date. This discovery has profound implications for the anticancer potential of niclosamide because SLC7A11 is considered as one of the promising drug targets for the treatment of not only TNBC, but also other cancers. Interestingly, the ability of niclosamide to interfere with the antioxidant machinery of TNBC cells does not stop with its direct effect as an inhibitor of SLC38A5 and SLC7A11. The drug also suppresses the expression of both transporters. We have not explored the signaling pathway that is responsible for this effect, but niclosamide is known to elicit its pharmacological effects by suppressing multiple signaling pathways, including Wnt, STAT3, mTOR, etc. Of note is the observation in the present study that niclosamide does interfere with mTOR signaling in TNBC cells. Additional studies are needed to tease out which of these pathways affected by niclosamide is responsible for the suppression of SLC38A5 and SLC7A11 in TNBC cells when exposed to the drug. It is possible that a single pathway may not be involved in the niclosamide-dependent regulation of SLC38A5 and SLC7A11. As multiple signaling pathways are affected by niclosamide, it is feasible that more than one signaling mechanism participate in the suppression of SLC38A5 and SLC7A11.

When the expression and function of SLC38A5 and SLC7A11 are impaired, one would expect decreased levels of glutathione and increased levels of lipid peroxidation in niclosamide-treated cells. This is indeed supported by the results of the experiments described in the present study. Niclosamide treatment decreases glutathione levels and increases iron and malondialdehyde levels in TNBC cells. In addition, niclosamide also decreases the levels of glutathione peroxidase 4 (GPX4) and ferritin (H chain). The exact mechanisms involved in this process remain to be investigated. It is known that glutathione peroxidase mRNA stability is influenced by the selenium nutritional status of the cells [44]. Selenium deficiency decreases the stability of GPX mRNAs via their 3′-untranslated regions. Therefore, we speculate that since the expression and function of SLC38A5 are drastically suppressed by niclosamide in TNBC cells, it would cause selenium deficiency due to impaired Se-Met delivery, which would then be expected to decrease the stability of GPX4 mRNA and hence its protein levels. Alternatively, any of the signaling pathways affected by niclosamide could also mediate the effect. The same is true for the decrease in ferritin levels. More work is needed to deduce the mechanisms involved in these effects.

The observed decrease in GPX4 and ferritin is directly related to the potentiating effect of niclosamide on ferroptosis in TNBC cells. The decrease in GPX4, coupled with the decrease in glutathione levels, would enhance lipid peroxidation, which is further supported by the observed increase in malondialdehyde levels. Ferritin sequesters iron and decreases the cellular levels of labile iron, thus protecting the cells from the potential pro-oxidant activity of free iron. Therefore, the decrease in ferritin levels in niclosamide-treated cells would be expected to increase the levels of labile iron, consequently enhancing lipid peroxidation and hence ferroptosis. Since tumor cells are known to be “addicted” to iron to promote their survival and proliferation, these cells potentiate their antioxidant machinery to protect themselves from ferroptosis. Our studies show that niclosamide effectively interferes with this protective mechanism, thus making the tumor cells susceptible to iron-induced cell death. This offers a novel anticancer mechanism for niclosamide.

The anticancer efficacy of niclosamide has been demonstrated in several cancers, both in vitro using cultured cells and in vivo using mouse xenografts [24,25,26]. The results of the present study offer further supportive evidence for the potential of niclosamide as an anticancer drug. Niclosamide suppresses the proliferation, colony formation and invasion/migration of TNBC cells. It also interferes with the growth of TNBC cells into tumors when xenografted in nude mice. However, a major hindrance in the successful use of this drug for cancer treatment appears to be the low bioavailability when the drug is given orally [45,46]. This does not necessarily negate the therapeutic potential of this drug. Studies are ongoing to increase the bioavailability of niclosamide either by changing the formulation or by using a prodrug approach. In the former, nanoformulations could enhance the bioavailability; in the latter, niclosamide can be structurally modified such that the modified drug is recognized as a transportable substrate for intestinal nutrient transporters (e.g., peptide transporter), thus increasing the oral bioavailability of the drug. Therefore, niclosamide holds great potential for the treatment of cancers and our present study provides strong support for its use in the treatment of TNBC. 

## Figures and Tables

**Figure 1 antioxidants-13-00291-f001:**
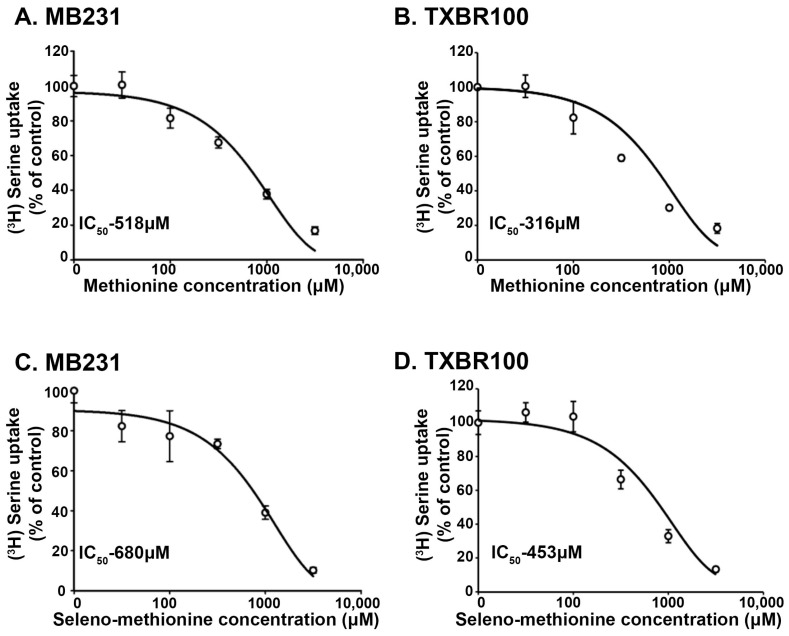
Dose–response relationship for the inhibition of SLC38A5-mediated serine uptake by methionine and selenomethionine in TNBC cell lines MB231 (**A**,**C**) and TXBR100 (**B**,**D**). The uptake of [^3^H]-serine that was mediated specifically by SLC38A5 was monitored, as detailed in the Section 2. Uptake in the absence of methionine or selenomethionine was taken as 100% and the uptake in the presence of methionine (**A**,**B**) or selenomethionine (**C**,**D**) was determined as the percent of this control uptake. The *IC*_50_ values (concentration of methionine or selenomethionine at which the inhibition was 50% of the control uptake) were calculated using the SigmaPlot 15.0 program (Grafiti, Palo Alto, CA, USA).

**Figure 2 antioxidants-13-00291-f002:**
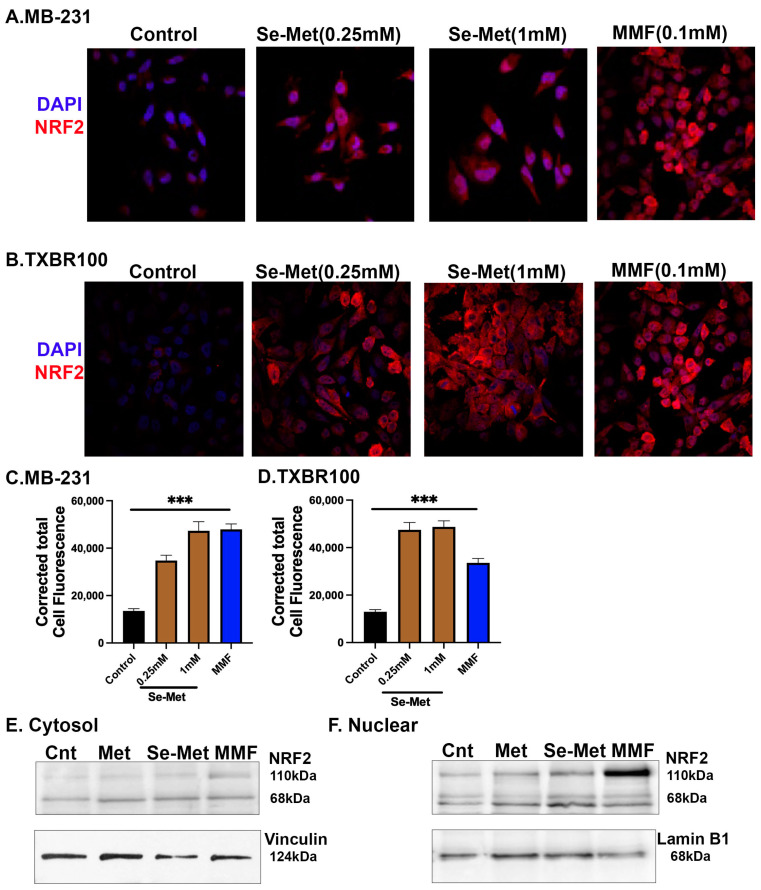
Effect of Se-Met on the expression levels of Nrf2 protein in the TNBC cell lines MB231 and TXBR100. The cells were treated with Se-Met at 0.25 mM or 1 mM for 24 h and then the cells were fixed and Nrf2 protein levels were monitored by immunofluorescence ((**A**,**C**), MB231; (**B**,**D**), TXBR100). Monomethylfumarate (MMF) was used at 0.1 mM as a positive control for induction of Nrf2 protein. DAPI was used as a marker for nucleus. The cells were also subjected to separation of cytosol and nuclear fractions which were then used for Western blot to determine Nrf2 protein levels (**E**,**F**). ***, *p* < 0.001.

**Figure 3 antioxidants-13-00291-f003:**
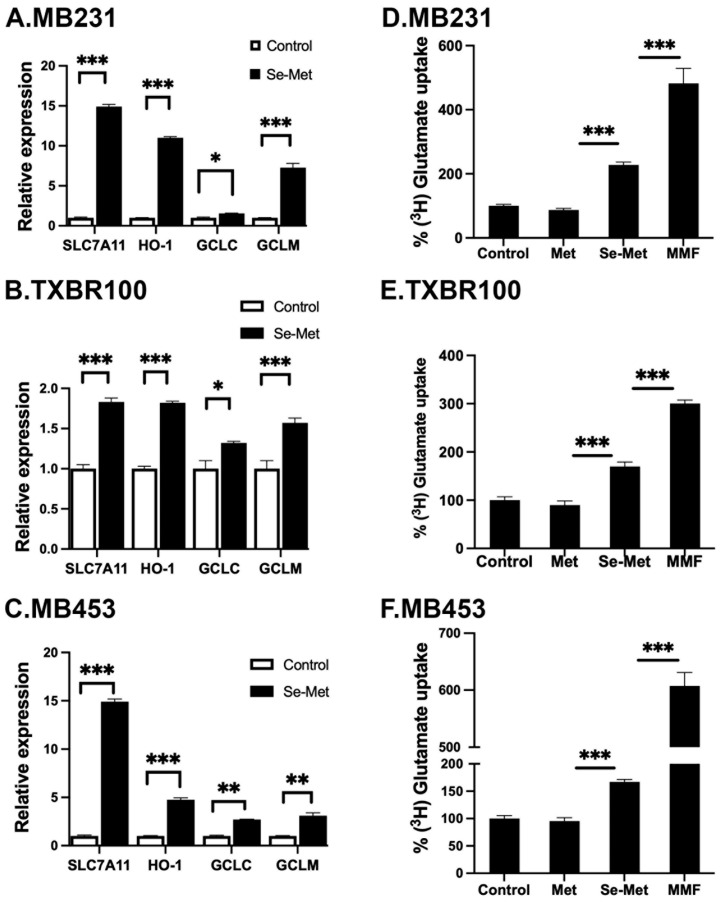
Influence of Se-Met on the expression of Nrf2 target genes in three different TNBC cell lines: (**A**,**D**) MB231; (**B**,**E**) TXBR100; (**C**,**F**) MB453. Cells were treated with Se-Met (1 mM) for 16 h, following which RNA was prepared from the cells for use in qRT-PCR. Cells cultured under identical conditions but in the absence of Se-Met were used as control. The mRNA levels for SLC7A11, heme oxygenase-1 (HO-1, catalytic subunit of glutamate-cysteine ligase (GCLC) and modulatory subunit of glutamate–cysteine ligase (GCLM)) were monitored by quantitative RT-PCR, as described in the Section 2. For each gene, the mRNA level in control (i.e., cultured in the absence of Se-Met) cells was taken as 1. Data are given as mean ± S.E. The functional activity of SLC7A11 was determined with glutamate as the transport substrate, as described in the Section 2. *, *p* < 0.05; **, *p* < 0.01; ***, *p* < 0.001.

**Figure 4 antioxidants-13-00291-f004:**
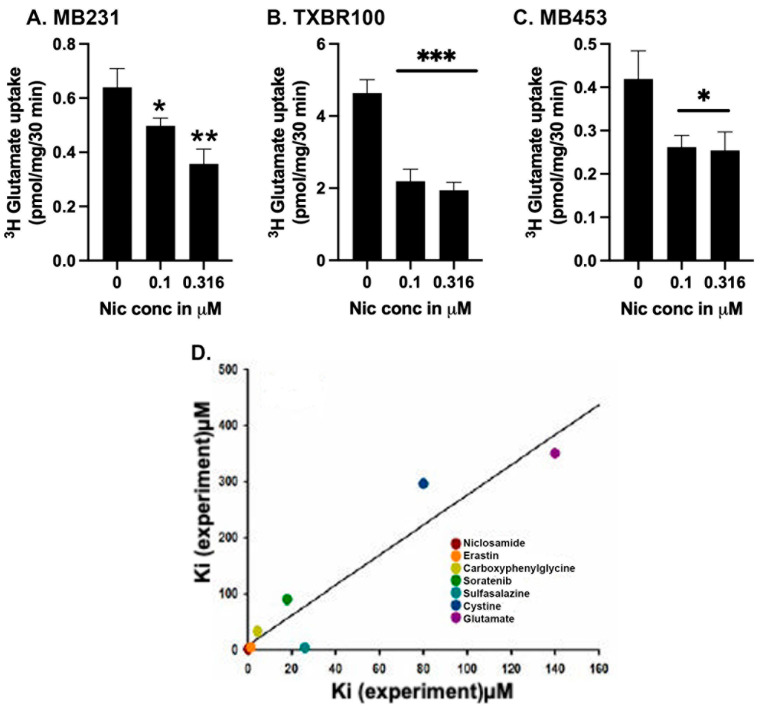
Direct effect of niclosamide on SLC7A11 transport activity (**A**–**C**) and the correlation between theoretically calculated and experimentally determined dissociation constants for various substrates and inhibitors (**D**). The transport activity of SLC7A11 was measured in three different TNBC cell lines (MB231, TXBR100 and MB453) as detailed in the Section 2. Niclosamide was present only during uptake. The dissociation constants for the substrates and inhibitors were determined in the present study or taken from published reports (see the text). For (**A**–**C**), data are given as mean ± S.E. *, *p* < 0.05; **, *p* < 0.01; ***, *p* < 0.001.

**Figure 5 antioxidants-13-00291-f005:**
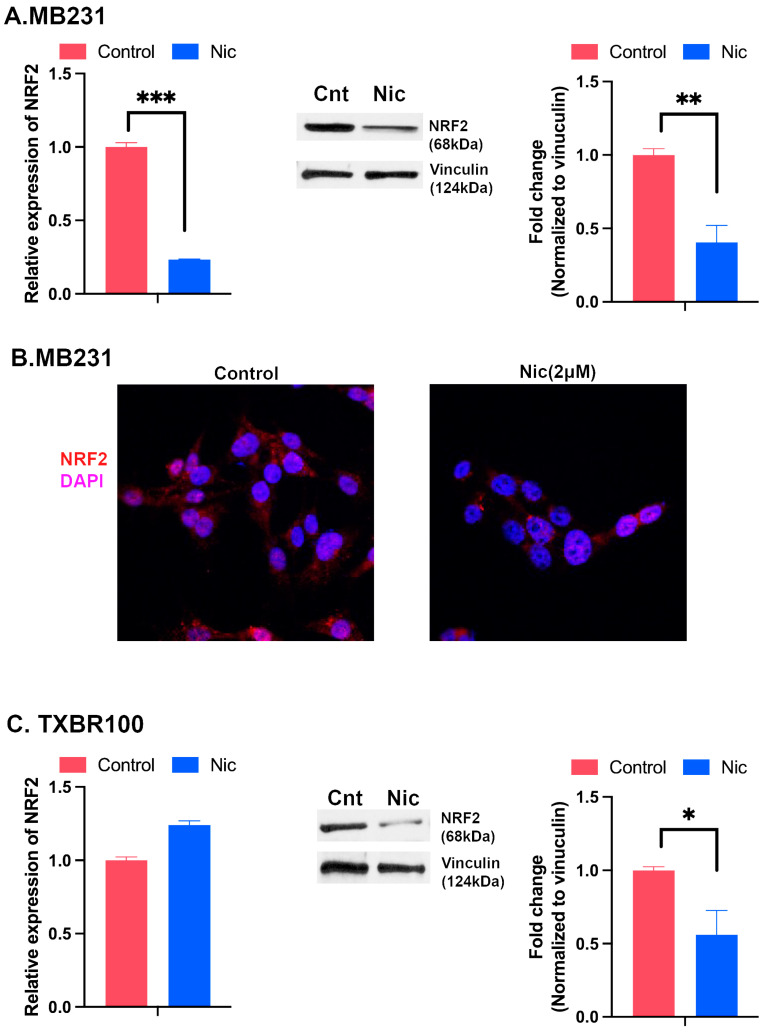
Influence of niclosamide on Nrf2 mRNA and protein expression in MB231 cells (**A**,**B**) and TXBR100 cells (**C**). The cells were treated with niclosamide (2 μM) for 24 h; RNA and protein lysates were then prepared from control and treated cells to monitor Nrf2 mRNA (left panel in (**A**,**C**)) and protein levels (right panels in (**A**,**C**)). mRNA levels were determined by qRT-PCR. Protein levels were determined by Western blot. Data are given as mean ± S.E. *, *p* < 0.05; **, *p* < 0.01; ***, *p* < 0.001.

**Figure 6 antioxidants-13-00291-f006:**
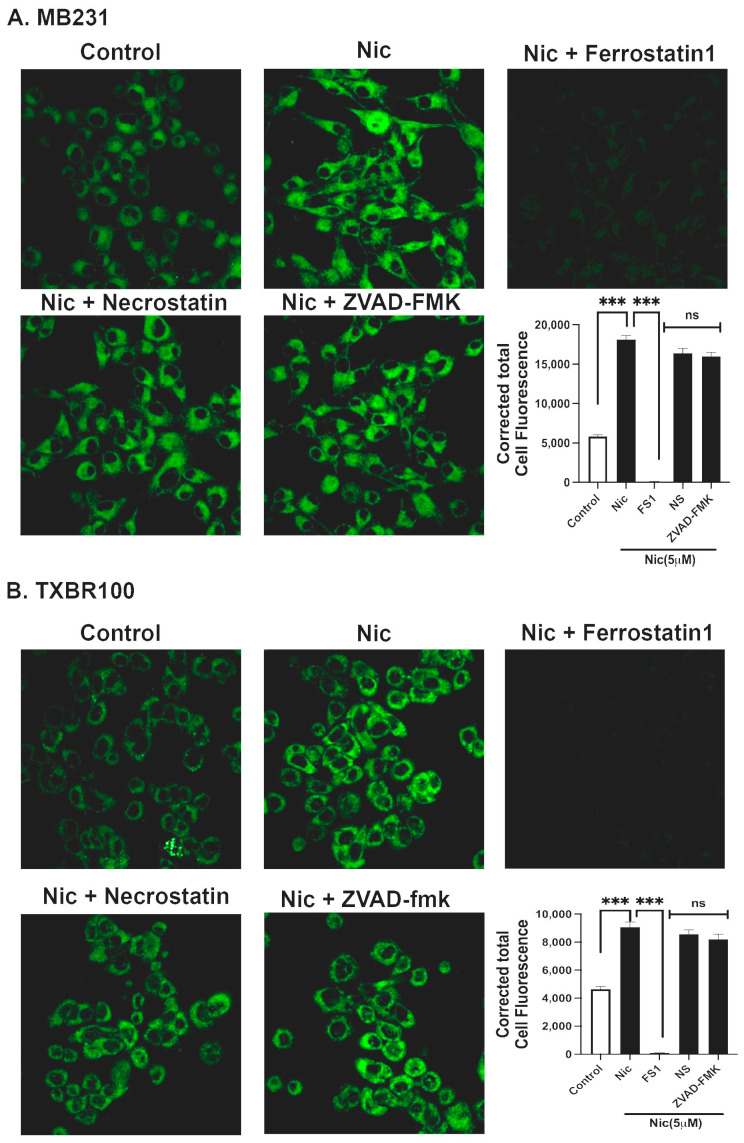
Induction of ferroptosis by niclosamide (5 μM) in MB231 cells and TXBR100 cells. Cells were treated with niclosamide (30 min treatment for MB231 cells; 2 h treatment for TXBR100 cells) with and without various cell death inhibitors (ferrostatin-1, necrostatin and ZVAD-fmk, each at 10 μM). Immunofluorescence images as well as quantification of the fluorescence signals are given. Data are given as mean ± S.E. ***, *p* < 0.001; ns, not significant.

**Figure 7 antioxidants-13-00291-f007:**
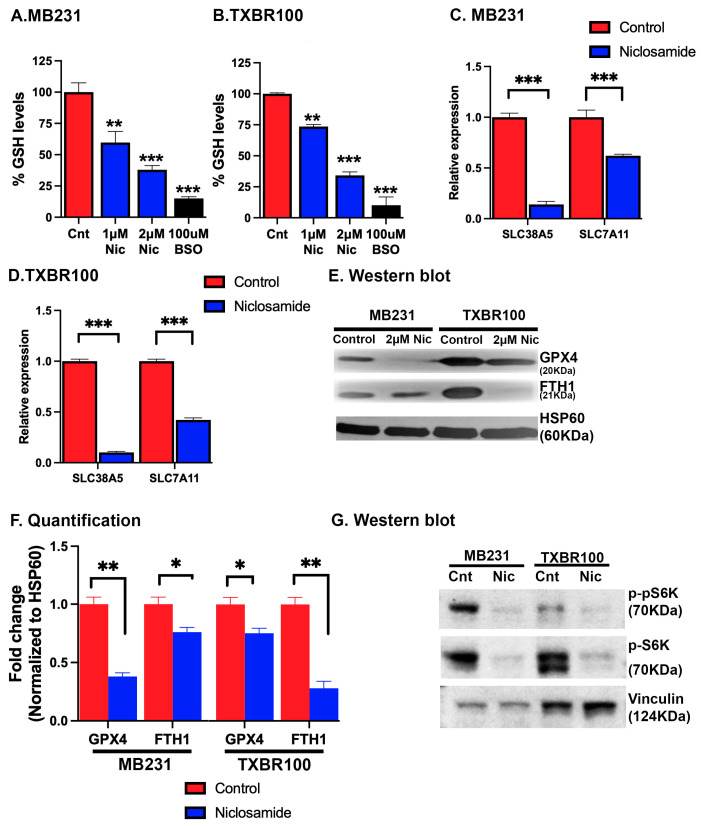
Induction of oxidative stress and suppression of SLC38A5 and SLC7A11 expression by niclosamide in TNBC cells. MB231 and TXBR100 cells were treated with niclosamide (2 μM) for 24 h and then RNA and protein lysates were prepared from untreated and treated cells. The levels of glutathione (**A**,**B**) were measured using a commercially available assay kit. The levels of mRNAs for SLC38A5 and SLC7A11 were monitored by qRT-PCR with 18S mRNA as the internal control (**C**,**D**). The protein levels of glutathione peroxidase 4 (GPX4) and ferritin heavy chain (FTH1) were analyzed by Western blot (**E**) and the protein bands quantified by densitometry (**F**). The protein levels of phospho-pS6K and pS6K were analyzed by Western blot (**G**). *, *p* < 0.05; **, *p* < 0.01; ***, *p* < 0.001.

**Figure 8 antioxidants-13-00291-f008:**
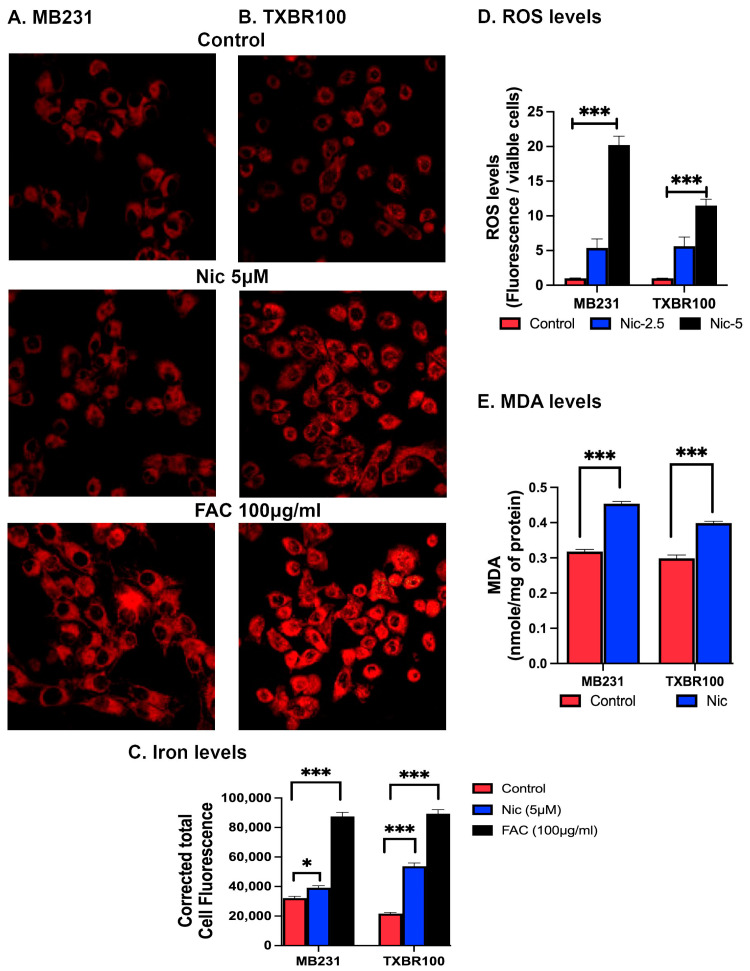
Effect of niclosamide on iron, intracellular reactive oxygen species (ROS) and lipid peroxidation. Cells were treated with niclosamide (5 μM) for 30 min and iron-specific fluorescence signals captured (**A**,**B**) and quantified (**C**) as described. Ferric ammonium citrate (FAC) was used as a source of iron (100 μg/mL) as a positive control to increase cellular levels of iron. MB231 and TXBR100 cells were treated with niclosamide (2.5 and/or 5 μM) for 24 h. Cells were then processed according to manufacturer’s protocols to quantify ROS or the lipid peroxdation marker malondialdehyde. (MDA). Data are given as mean ± S.E. *, *p* < 0.05; ***, *p* < 0.001.

**Figure 9 antioxidants-13-00291-f009:**
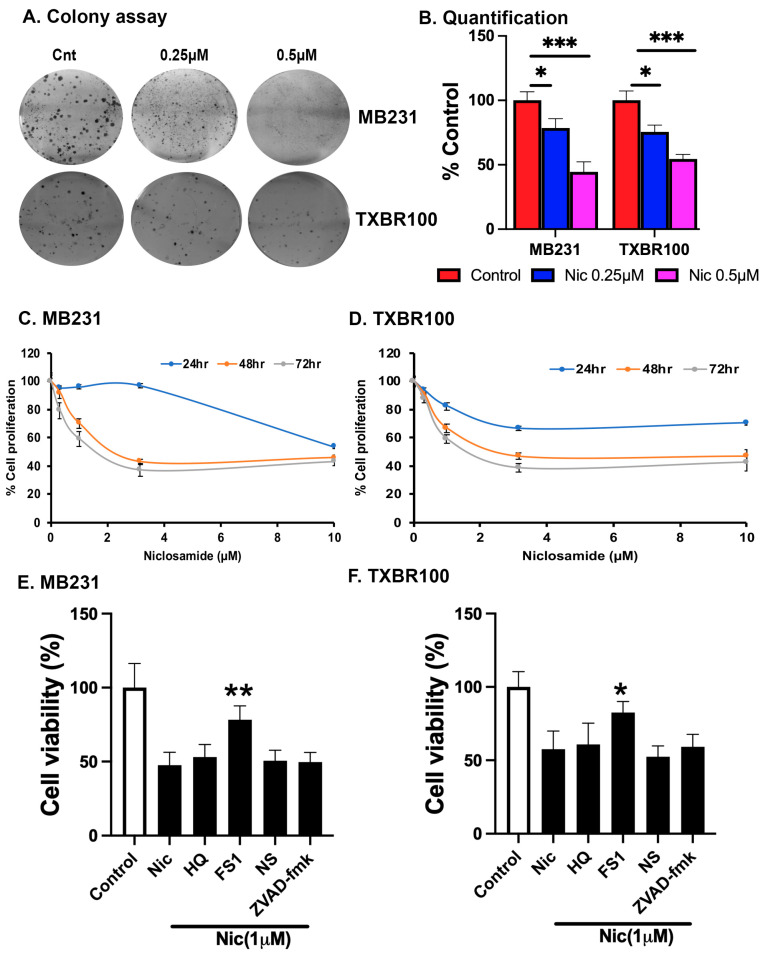
Effects of niclosamide on colony formation (**A**,**B**) and cell proliferation (**C**,**D**). Effects of different cell death inhibitors on niclosamide-induced cytotoxicity (**E**,**F**). Cells were treated with 1 μM of niclosamide with or without hydroxychloroquine (HQ) (25 μM) (an inhibitor of autophagic cell death), ferrostatin-1 (10 μM) (an inhibitor of ferroptosis), necrostatin (10 μM) (an inhibitor of necroptosis) and ZVAD-fmk (10 μM) (an inhibitor of apoptosis) for 48 h and then MTT assay was performed. Cell viability was calculated as percent of control cells. *, *p* < 0.05; **, *p* < 0.01; ***, *p* < 0.001.

**Figure 10 antioxidants-13-00291-f010:**
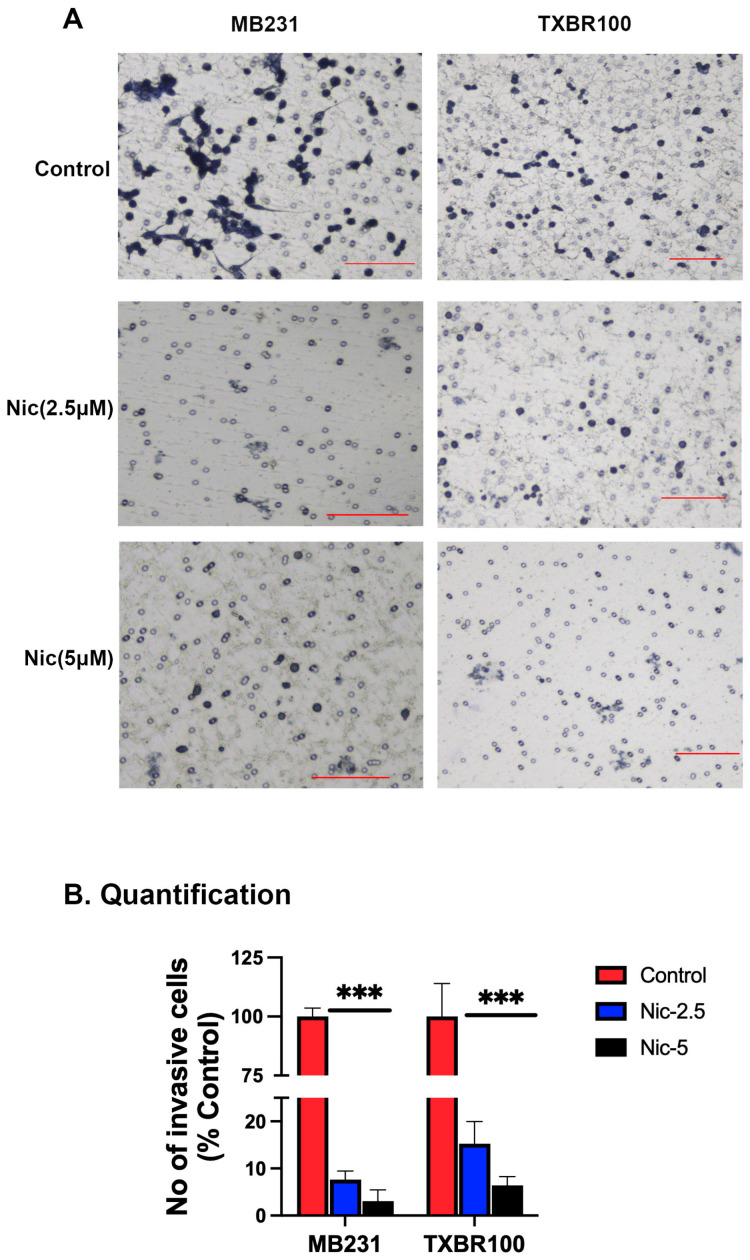
Effect of niclosamide on TNBC cells on their ability to invade and migrate across Matrigel. MB231 and TXBR100 cells were treated with vehicle control or niclosamide (2.5 µM and 5 µM) for 24 h in Matrigel-coated transwell chambers. The invasive cells that moved to the other side of the membrane were stained with crystal violet and then photographed (10×; scale bar, 100 μm) (**A**) and also counted (**B**) under microscope. Data for the invaded cells are given as percent of total number of cells originally seeded on the top side of the membrane. ***, *p* < 0.001.

**Figure 11 antioxidants-13-00291-f011:**
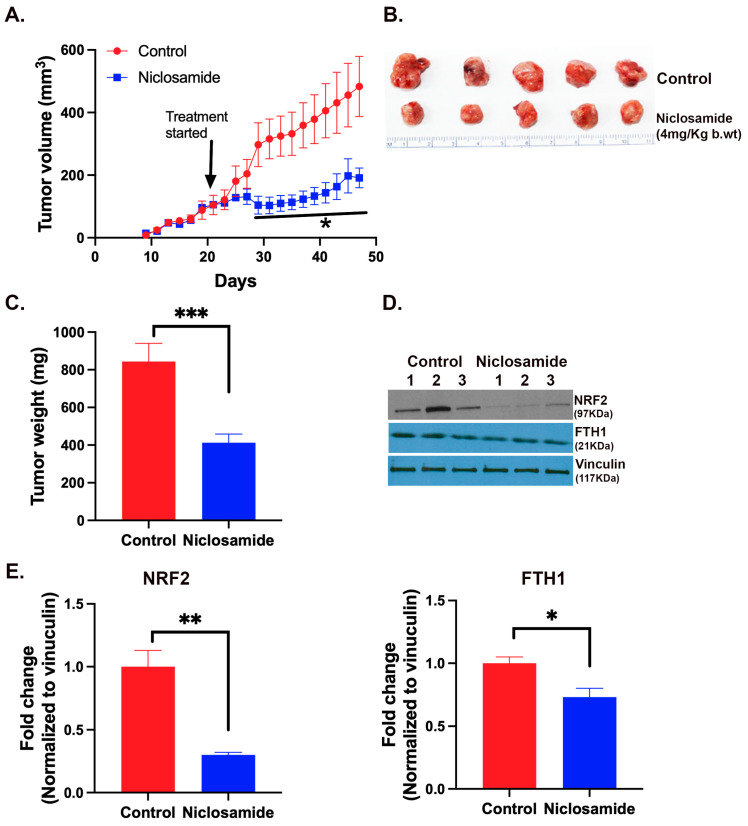
Tumor-suppressive efficacy of niclosamide in vivo in mouse xenograft. MB231 cells were injected into mammary fat pad of nude mice and the tumors were allowed to grow to an approximate size of 100–150 mm^3^. Niclosamide administration began at this point in one group of mice while the other group was left untreated. The drug was given intraperitoneally daily (4 mg/kg body weight) and the treatment continued for 4 weeks. Tumor volume during the experimental period (**A**) and tumor size and weights at the end of the experimental period (**B**,**C**) are shown. Tumor tissues were prepared for Western blot to monitor the protein levels of Nrf2 and ferritin heavy chain (FTH1) (**D**,**E**). *, *p* < 0.05; **, *p* < 0.01; ***, *p* < 0.001.

## Data Availability

All the data related to this study are given in the manuscript and in the Appendix A, which is freely available to the scientific community and the public.

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
