# Peer review of "Induction of Oxidative Stress and Ferroptosis in Triple-Negative Breast Cancer Cells by Niclosamide via Blockade of the Function and Expression of SLC38A5 and SLC7A11"

_antioxidants, 2024, doi:10.3390/antiox13030291_

Round 1

Reviewer 1 Report (Previous Reviewer 3)

The authors have responded to all the queries from the reviewer with proper explanation. The reviewer opines that the paper is now ready for publication in its revised form.

The findings from the paper can contribute in understanding the mechanism of niclosamide against TNBC and its future application in clinical trials. 

Author Response

Thank you.

Reviewer 2 Report (Previous Reviewer 2)

The author has addressed the concerns and made careful revisions to the original text, resulting in a greatly improved current version. I have no further comments.

I have no further comments.

Author Response

Thank you.

Reviewer 3 Report (Previous Reviewer 1)

None.

Second review on the manuscript of Mathew, M. et al.: “Induction of oxidative stress and ferroptosis in triple-negative breast cancer cells by niclosamide via blockade of the function and expression of SLC38A5 and SLC7A11”.

In this manuscript, Authors explore how cancer cells improve their antioxidant capacity to deal with the higher levels of iron required for their accelerated metabolic activity. In addition, Authors also explore the mechanism involved in the oxidative stress and ferroptosis induced by niclosamide in triple-negative breast cancer cells. This could have great implications for cancer treatment.

The corresponds to a second version of the manuscript after peer-review. Authors have made substantial alterations to the original version of the manuscript, in line with the Reviewer’s suggestions. So, I congratulate Authors for the good job.

Author Response

Thank you very much for your kind words.

This manuscript is a resubmission of an earlier submission. The following is a list of the peer review reports and author responses from that submission.

Round 1

Reviewer 1 Report

Comments and Suggestions for Authors

Review on the manuscript of Mathew, M. et al.: “Induction of oxidative stress and ferroptosis in triple-negative breast cancer cells by niclosamide via blockade of the function and expression of SLC38A5 and SLC7A11”.

In this manuscript, Authors explore how cancer cells improve their antioxidant capacity to deal with the higher levels of iron required for their accelerated metabolic activity. In addition, Authors also explore the mechanism involved in the oxidative stress and ferroptosis induced by niclosamide in triple-negative breast cancer cells. This could have great implications for cancer treatment.

The manuscript is easy to follow and on a very important research topic. The work is very elegant, and, in general, the experiments were well done. It was great pleasure to me reading this work. Thus, the issues that arise to me are listed below, so, I hope Authors find the following comments and suggestions useful.

1 – Authors show that exposure of the TNBC cell line MB231 to Se-Met increased cellular levels as well as nuclear localization of Nrf2 (Figure 2). I recommend Authors to make some quantifications of Nrf2 levels (and show a graph with this data), to clearly show an increase in Nrf2 expression levels (like Authors did in Figure 6, where Nrf2 expression was quantified from WB and qRT-PCR experiments).

2 – Authors show that, in support of increased Nr2 expression, the transport activity of SLC7A11 increased substantially in Se-Met-treated cells, but not in Met-treated cells. Therefore, this suggests that methionine is not able to induce changes in Nrf2 expression in these cell lines. As a proof-of-concept, I recommend Authors to show this experiment in the manuscript (I imagine Authors have this data).

3 – Authors report a significant difference in the susceptibility to niclosamide-induced ferroptosis between MB231 cells and TXBR100 cells. Could this phenotype be related to differences in the expression of the cystine transporter SLC7A11 or the Nrf2 response between cell lines? Can Authors elaborate a little bit more on this topic?

4 – Based on the data that SLC38A5-dependent transport of Se-Met increases the antioxidant fitness of cancer cells, in cases of cancer, is it recommend having a diet poor in Se-Met? Can Authors elaborate a little bit on it in the discussion section?

Author Response

  1. We have now quantified the fluorescence signals and included the data in Fig. 2 C & D.
  2. In response to the comment by another reviewer, we have now isolated the cytosolic and nuclear fractions and then performed western blot for Nrf2. We did this experiment for control cells and cells treated with methionine, Se-Met and monomethylfumarate. The data are given in Fig. 2 E & F. This experiment shows that methionine did not increase the levels of Nrf2 protein, neither in the cytosol nor in the nucleus, in contrast to Se-Met and the positive control monomethylfumarate.
  3. We do not know the reasons for the difference between MB231 and TXBR cells in terms of susceptibility to niclosamide-induced ferroptosis. Nonetheless, the difference is only in the dose and time-dependence. TXBR100 cells do respond to niclosamide and undergo ferroptosis when exposed to niclosamide.
  4. We do not want to speculate on this topic because strategies to increase oxidative stress in cancer patients is like a two-edged sword. Yes, increasing oxidative stress with diets deficient in Se-Met may kill cancer cells, but the same strategy may also have a facilitating effect on carcinogenesis; oxidative stress can cause mutations and promote cancer initiation. We sincerely hope that the reviewer would agree with our argument.

Reviewer 2 Report

Comments and Suggestions for Authors

Marilyn Mathew and colleagues examined how niclosamide triggers oxidative stress and ferroptosis in triple-negative breast cancer cells by hampering the operation and manifestation of SLC38A5 and SLC7A11. This article only provides a phenotypic functional result and does not go into the underlying mechanism in detail.

1. To ascertain whether niclosamide-induced triple-negative breast cancer is ferroptosis, it is necessary to detect iron ion content and lipid ROS, as well as to combine various cell death inhibitors.

2. The presentation of the results of the article is not visually appealing, since the results of the same topic are not grouped together in one figure, but are instead shown as separate pictures, giving the impression of insufficient effort.

3. Immunofluorescence is not the only way to detect NRF2; Western Blotting is also necessary to measure the levels of both cytoplasmic and nuclear proteins.

4. Without internal references, it is not possible to evaluate the accuracy of the WB results.

Author Response

  1. We consider this as an elaborate study to functionally link the two amino acid transporters, SLC38A5 and SLC7A11, and explore their potential as drug targets for cancer therapy. The study does provide mechanistic insight into this topic. For example, in the case of niclosamide-induced ferroptosis in TNBC cells, we have demonstrated the increase in iron levels, increase in ROS levels, increase in malondialdehyde levels, decrease in glutathione levels and decrease in GPX4 levels. All of these data are mechanistically relevant to ferroptosis. In the mouse xenograft experiment, we link the decrease in tumor growth in response to niclosamide treatment to decreased expression of the two amino acid transporters. We sincerely hope that the reviewer would agree now that the newly added data in the revised version addresses his concern satisfactorily.
  2. We are genuinely surprised at this comment. We spent a considerable amount of time organizing the order of presentation of the data so that there is a logic and rationale for the sequence of data presentation. After seeing the reviewer’s comment, we went back and carefully looked at the order of data presentation, and again found the order to be logical: (i) transport of Se-Met by SLC38A5; (ii) effect of Se-Met on Nrf2; (iii) influence of Se-Met on expression of antioxidant proteins, including SLC7A11; (iv) inhibition of SLC7A11 by niclosamide; (v) effect of niclosamide on ferroptosis and related components; (vi) in vivo effect of niclosamide on tumor growth.
  3. We have now isolated the cytoplasmic and nuclear fractions and performed western blot for Nrf2. The data are shown in Fig. 2 E & F. Thank you for the suggestion.
  4. All western blots included internal controls (vinculin, Lamin B1, and HSP60 depending on whether the experiment is related to cytosolic fraction, nuclear fraction, or total lysate).

Reviewer 3 Report

Comments and Suggestions for Authors

The article by Ganapathy et al. describes the anticancer activity of an antiparasitic drug niclosamide in TNBC via targeting intracellular antioxidant machineries through the interaction with two amino acid transporters. This article highlights the functional coupling between amino acid transporters that are often upregulated in cancer cells and investigated as emerging therapeutic targets. Niclosamide has been reported to evoke antitumor activity in several cancer cell lines, but the mode of action is still a topic of interest which has been investigated carefully by the authors. The reviewer recommends publication of this article after following a revision. The reviewer has several suggestion and queries as described below:

1)     Figure 10 repeatedly mention vinuculin in the several experiments without any mention of it in the related section. The authors should clarify.

2)    The reviewer is curious if niclosamide can also accentuate the level of intracellular ROS as the main mode of action is related to generating oxidative stress. The authors should investigate this and mention the result in the draft.

3)    Similarly, the authors should also look into the inhibition of mTOR signaling pathway by niclosamide and incorporate the results in the current draft.

4)    Metastasis is one of the culprits for turning TNBC as a deadly disease. The reviewer would like to know if niclosamide can show promises in inhibiting metastasis of this aggressive form of cancer.

Author Response

  1. Vinculin is just an internal control for protein normalization.
  2. We have now measured ROS levels in control and niclosamide-treated cells. The data are provided in Fig. 6D.
  3. We have now probed mTOR signaling pathway, and the data are shown in Fig. 7G.
  4. We have not yet explored the potential influence of niclosamide on metastasis of TNBC. It is something we can investigate in the future. However, we have now studied the influence of niclosamide on invasion/migration property of TNBC cells and the data show that the drug does suppress this characteristic. This could be taken as evidence that niclosamide might have the ability to suppress metastasis of TNBC. Additional investigations are needed to address this topic in depth.